# The Apoptotic and Anti-Warburg Effects of Brassinin in PC-3 Cells via Reactive Oxygen Species Production and the Inhibition of the c-Myc, SIRT1, and β-Catenin Signaling Axis

**DOI:** 10.3390/ijms241813912

**Published:** 2023-09-10

**Authors:** Hyeon Hee Kwon, Chi-Hoon Ahn, Hyo-Jung Lee, Deok Yong Sim, Ji Eon Park, Su-Yeon Park, Bonglee Kim, Bum-Sang Shim, Sung-Hoon Kim

**Affiliations:** College of Korean Medicine, Kyung Hee University, Seoul 02447, Republic of Korea; dwfqwd223@hanmail.net (H.H.K.); ach2565@naver.com (C.-H.A.); hyonice77@naver.com (H.-J.L.); simdy0821@naver.com (D.Y.S.); wdnk77@naver.com (J.E.P.); waterlilypark@naver.com (S.-Y.P.); bongleekim@khu.ac.kr (B.K.); eshimbs@khu.ac.kr (B.-S.S.)

**Keywords:** prostate cancer, Brassinin, glycolysis, ROS, SIRT1, c-Myc, β-catenin

## Abstract

Though Brassinin is known to have antiangiogenic, anti-inflammatory, and antitumor effects in colon, prostate, breast, lung, and liver cancers, the underlying antitumor mechanism of Brassinin is not fully understood so far. Hence, in the current study, the apoptotic mechanism of Brassinin was explored in prostate cancer. Herein, Brassinin significantly increased the cytotoxicity and reduced the expressions of pro-Poly ADP-ribose polymerase (PARP), pro-caspase 3, and B-cell lymphoma 2 (Bcl-2) in PC-3 cells compared to DU145 and LNCaP cells. Consistently, Brassinin reduced the number of colonies and increased the sub-G1 population and terminal deoxynucleotidyl transferase (TdT) dUTP Nick-End Labeling (TUNEL)-positive cells in the PC-3 cells. Of note, Brassinin suppressed the expressions of pyruvate kinase-M2 (PKM2), glucose transporter 1 (GLUT1), hexokinase 2 (HK2), and lactate dehydrogenase (LDH) as glycolytic proteins in the PC-3 cells. Furthermore, Brassinin significantly reduced the expressions of SIRT1, c-Myc, and β-catenin in the PC-3 cells and also disrupted the binding of SIRT1 with β-catenin, along with a protein–protein interaction (PPI) score of 0.879 and spearman’s correlation coefficient of 0.47 being observed between SIRT1 and β-catenin. Of note, Brassinin significantly increased the reactive oxygen species (ROS) generation in the PC-3 cells. Conversely, ROS scavenger NAC reversed the ability of Brassinin to attenuate pro-PARP, pro-Caspase3, SIRT1, and β-catenin in the PC-3 cells. Taken together, these findings support evidence that Brassinin induces apoptosis via the ROS-mediated inhibition of SIRT1, c-Myc, β-catenin, and glycolysis proteins as a potent anticancer candidate.

## 1. Introduction

Prostate cancer is the second most common carcinoma in men all over the world [1,2]. According to 2020 WHO data on cancer incidence and mortality statistics, its new incidence and mortality were 414,259 (7.3%) and 375,304 (3.8%), respectively [3]. Despite current therapeutic methods such as surgery, chemotherapy, radiotherapy, immunotherapy, androgen deprivation therapy (ADT), and hormonal manipulation by antiandrogens such as bicalutamide, nilutamide, ketoconazole, or corticosteroids [4,5], the new case incidence and mortality death of prostate cancer are still increasing rapidly. Recently, some target therapies for prostate cancer have gained attention, including those targeting the prostate-specific membrane antigen (PSMA) [6], epidermal growth factor receptor (EGFR) [7], vascular endothelial growth factor (VEGF) [8], and BCL2-associated athanogene 1 L (BAG1L) [9].

It is well documented that the Warburg effect enhances the consumption of glucose in several cancers compared to normal tissues under accelerated aerobic glycolysis [10]. Indeed, the Warburg effect is critically involved in prostate cancer progression [11], with Phosphatase and Tensin Homolog deleted on Chromosome 10 (PTEN) loss [12] and glycolysis-associated proteins such as pyruvate kinase-M2 (PKM2), glucose transporter 1 (GLUT1), hexokinase 2 (HK2), and lactate dehydrogenase (LDH) [13,14]. Hence, the inhibition of the Warburg effect is regarded as a therapeutic strategy in prostate cancer therapy [11,15].

Interestingly, c-Myc, one of the Myc family, known as oncogenes [16,17], regulates glucose and glutamine metabolism in several cancers, including prostate cancer [18,19]. In addition, silent information regulator 1 (SIRT1) enhances tumorigenesis and metastasis via the deacetylation of tumor suppressors [20] and also increases glucolipid metabolic conversion to facilitate cancer progression [21]. Likewise, WNT/β-catenin signaling promotes self-renewal or expansion in prostate cancer stem cells [22] and modulates aerobic glycolysis [23,24].

Furthermore, it is noteworthy that the Warburg effect enables cancer cells to avoid excess reactive oxygen species (ROS) generation by limiting the pyruvate flux into mitochondrial oxidative metabolism [25], since the excessive production of ROS can initiate programmed cell death in several cancers [26].

Recently, some natural compounds such as curcumin [27], sulforaphane [28], quercetin [29], diallyl trisulfide [30], alternol [31], silibinin [32], and decursin [33] have been determined to be of interest for treating prostate cancers because they modulate oxidative stress and other signaling pathways. Similarly, Brassinin, a phytoalexin in cruciferous vegetables, has been reported to exert anti-inflammatory [34], anti-proliferative [35], and anticancer effects in colon [36], prostate [37], breast [38], lung [39], and liver [40] cancers. Nonetheless, the underlying anticancer mechanism of Brassinin is not yet fully understood in prostate cancer. Thus, in the current study, the apoptotic mechanism of Brassinin was explored in prostate cancers in association with glycolysis-related proteins and the c-Myc/SIRT1/β-catenin signaling axis.

## 2. Results

### 2.1. Cytotoxic Effect of Brassinin in Human Prostate Cancer Cells

To estimate the cytotoxic effect of Brassinin (Figure 1a), a cell viability assay was performed in PC-3, DU145, and LNCaP prostate cancer cells and an MTT assay was carried out using human prostatic epithelial cell line RWPE-1 cells. The cells were treated with indicated concentrations of Brassinin (0, 40, 60, 80, 120, and 160 μM) for 48 h. Herein, Brassinin significantly suppressed the viability in the PC-3 cells (Figure 1b), while the DU145 and LNCaP cells were not sensitive to Brassinin compared to the PC-3 cells, and it did not hurt the RWPE-1 cells. Furthermore, Brassinin significantly reduced the number of colonies in the PC-3 cells, as seen in a colony formation assay (Figure 1c).

### 2.2. Brassinin Induced Apoptosis in PC-3 Cells

To confirm the apoptotic effect of Brassinin, a cell cycle assay and Western blotting were performed on PC-3, DU145, and LNCaP cells treated by Brassinin. Here, Brassinin reduced the expressions of pro-Caspase3, pro-PARP, and Bcl-2 in the PC-3 cells compared to the DU145 and LNCaP cells (Figure 2a) and enhanced the sub-G1 population in the PC-3 cells (Figure 2b). Consistently, Brassinin increased the number of TUNEL-positive cells compared to the untreated control, as seen in an FACS analysis (Figure 2c).

### 2.3. Brassinin Effectively Suppressed the Expression of Glycolysis-Related Proteins in PC-3 Cells

To confirm whether the apoptotic effect of Brassinin was related to glycolysis, Western blotting was carried out on the Brassinin-treated PC-3 cells. As shown in Figure 3, Brassinin suppressed the expressions of HK2, Glut1, PKM2, and LDH as glycolytic proteins compared to the untreated control in the PC-3 cells.

### 2.4. Brassinin Reduces the Expression of SIRT1, β-Catenin, and c-Myc in PC-3 Cells

According to RNA sequencing data from the TCGA and GTEx databases using GEPIA, though the significant difference in SIRT 1 was not accepted between cancer patients (n = 492) and a normal group (n = 152) (Figure 4a), a lower SIRT1 expression was associated with an improved overall survival in prostate cancer patients (Figure 4b). To confirm whether or not the anti-glycolytic effect of Brassinin is mediated by the SIRT1 and β-catenin signaling axis, Western blotting was performed on the Brassinin-treated PC-3 cells. Brassinin decreased the expressions of SIRT1, β-catenin, and c-Myc in the PC-3 cells (Figure 4c).

### 2.5. Effect of Brassinin on Interaction between SIRT1 and β-Catenin in PC-3 Cells

The score of the protein–protein interaction (PPI) between SIRT1 and β-catenin was 0.879 (Figure 5a), while the close correlation between SIRT1 and β-catenin in terms of the spearman’s correlation coefficient was 0.47, as seen in the cBioportal correlation database (Figure 5b). To confirm the interaction between SIRT1 and β-catenin, immunoprecipitation was conducted in the PC-3 cells. IP revealed that Brassinin disrupted the binding of SIRT1 and β-catenin after the treatment with Brassinin for 48 h in the PC-3 cells (Figure 5c).

### 2.6. Brassinin Increased ROS Production in PC-3 Cells

To examine the role of ROS in the Brassinin-induced apoptosis in the PC-3 cells, a flow cytometry analysis was carried out using DCFH-DA dye. Brassinin significantly increased the ROS production at concentrations of 60 and 80 μM compared to the untreated control (44.20%) in the PC-3 cells (Figure 6). However, there was no significant difference for the ROS production between using 60 and 80 µM of Brassinin, though ROS were generated more at 60 μΜ (53.52%) than at 80 μΜ (52.89%), different from the cytotoxicity and colony formation assays’ data.

### 2.7. ROS Inhibitor N-Acetyl-L-Cysteine (NAC), Disturbs Brassinin-Induced Apoptosis in PC-3 Cells

ROS scavenger NAC reduced the ability of Brassinin to generate ROS production in the PC-3 cells, while Brassinin increased the ROS production in the PC-3 cells (Figure 7a). Additionally, NAC interfered with the capacity of Brassinin to attenuate the expressions of SIRT1, β-catenin, pro-Caspase3, and pro-PARP in the PC-3 cells (Figure 7b).

## 3. Discussion

In the present work, the underlying antitumor mechanism of Brassinin was explored in prostate cancer cells in association with glycolysis and the c-Myc/SIRT1/β-catenin signaling axis. Brassinin, derived from Chinese cabbage, exerts an antitumor effect in several cancers including colon [36], prostate [37], breast [38], lung [39], and liver [40]. Notably, although the Brassinin-induced apoptosis in the PC-3 cells was found to occur via the inhibition of the PI3K/Akt/mTOR/S6K1 signaling axis [41], there has been no evidence for the role of glycolysis-related proteins and c-Myc/SIRT1/β-catenin signaling in the Brassinin-induced antitumor effect in prostate cancer to date. Here, Brassinin significantly exerted cytotoxicity in PC-3, DU145, and LNCaP prostate cancer cells, without hurting normal prostatic epithelial RWPE-1 cells, implying that androgen-independent PC-3 cells are more susceptible to Brassinin compared to DU145 and androgen-dependent LNCaP cells. PC-3 is a cell line from grade IV adenocarcinoma with a high metastatic potential and DU145 is a cell line from prostate carcinoma with a moderated metastatic potential [42]. Morphologically, the DU-146 cell line looks to proliferate more rapidly compared to the PC-3 cell line [43], but Brassinin inhibits the proliferation of PC-3 cells, rather than the more proliferative DU145 or androgen-sensitive LNCaP cells. Likewise, Kim et al. [41] reported that Brassinin reduced the constitutive phosphorylation of Akt against androgen-independent PC-3 cells compared to androgen-dependent LNCaP cells. Consistently, Brassinin alleviated the expressions of pro-PARP, pro-caspase 3, and BCL2 in the PC-3 cells compared to the DU145 and LNCaP cells, implying an apoptotic effect of Brassinin, since the cleavages of PARP and caspase3 are a known feature of caspase-dependent apoptosis [44].

In particular, it is noteworthy that the LNCaP cells were more resistant to the cytotoxic and apoptotic effects of Brassinin, which should be further explored in vitro and in vivo. Consistently, Brassinin increased the sub-G1 population and number of TUNEL-positive cells for the apoptosis portion in the PC-3 cells, indicating the apoptotic potential of Brassinin in PC-3 cells, since internucleosomal DNA fragmentation indicates a hypoploid cell population commonly called the “Sub-G1” phase [45]. Furthermore, Brassinin reduced the number of colonies, since a clonogenic assay or colony formation assay is an in vitro cell survival assay based on the ability of a single cell to grow into a colony, implying the anti-proliferative activity of Brassinin. However, the antiproliferative activity of Brassinin was better than its cytotoxic effect, implying that the long-term effect of Brassinin is better than its short-term cytotoxic effect, with the necessity of further mechanistic studies.

It is well documented that aerobic-glycolysis-related proteins are overexpressed in cancer cells more than in normal cells [10], since glucose is converted into pyruvate, followed by lactate formation under aerobic glycolysis, which is called the Warburg effect [46]. In general, glycolysis-related proteins are known as glucose transporter 1 (GLUT1), hexokinase 2 (HK2), pyruvate kinase-M2 splice isoform (PKM2), c-Myc, lactate dehydrogenase A (LDH-A), p-PDK1/PDK1 (pyruvate dehydrogenase kinase 1), and Caveolin-1. In our work, Brassinin suppressed the expressions of PKM2, Glut1, HK2, and LDH in the PC-3 cells, demonstrating the anti-Warburg potential of Brassinin due to its better antiproliferative effect compared to its cytotoxicity effect. Accumulating evidence reveals that cMyc promotes glycolysis and oxidative phosphorylation in several cancers [47]. In addition, Chen and his colleagues reported that SIRT1 promotes hexokinase-2 (HK-2) as a strong oncogenic driver to enhance glycolysis and tumorigenesis [48]. Furthermore, Wnt/β-catenin signaling can promote cancer cell glycolysis in association with c-Myc [49,50]. Herein, Brassinin significantly reduced the expressions of SIRT1, c-Myc, and β-catenin in the PC-3 cells and also disrupted the binding of SIRT1 with β-catenin, along with a protein–protein interaction (PPI) score of 0.879 and a spearman’s correlation coefficient of 0.47 between the SIRT1 and β-catenin, indicating an important role of c-Myc/SIRT1/β-catenin in the anti-Warburg effect of Brassinin. Indeed, Zhou et al. [51] reported that SIRT1 directly binds to and deacetylates β-catenin and increases its accumulation in the nuclei of C3H10T1/2 mesenchymal stem cells.

Emerging evidence supports that reactive oxygen species (ROS) activate pro-tumorigenic signaling by promoting cell survival and proliferation and driving DNA damage and genetic instability [52]. Additionally, boosting lipid catabolism, which has a pro-tumoral effect, is dependent on ROS production and HIF1α induction in cervical cancer cells [53]. Of note, Brassinin significantly increased the ROS generation in the PC-3 cells, though this was not concentration dependent. Conversely, the ROS inhibitor NAC reversed the ability of Brassinin to attenuate pro-PARP, pro-Caspase3, SIRT1, and β-catenin in the PC-3 cells, strongly demonstrating the critical role of ROS in the antitumor effect of Brassinin.

In summary, Brassinin significantly enhanced the cytotoxicity and attenuated the number of colonies and expressions of pro-caspase 3, pro-PARP, and Bcl-2 compared to the DU145 and LNCaP cells, without hurting normal RWPE-1 cells. Furthermore, it increased the sub-G1 population and TUNEL-positive cells in the PC-3 cells. Of note, Brassinin suppressed the expressions of Glut1, HK2, PKM2, LDH, SIRT1, c-Myc, and β-catenin in the PC-3 cells. Furthermore, Brassinin disrupted the binding of SIRT1 with β-catenin and increased ROS production, while NAC reversed the ability of Brassinin to alleviate the expressions of pro-PARP, pro-Caspase3, SIRT1, and β-catenin in the PC-3 cells.

## 4. Materials and Methods

### 4.1. Brassinin Preparation

Brassinin was purchased from Sigma–Aldrich (Sigma, St. Louis, MO, USA) and stock solution (100 mM) was prepared in sterile DMSO and stored at −80 °C until the experiments were conducted.

### 4.2. Cell Culture

The human prostate cancer DU145, PC-3, and LNCaP cells were obtained from the Korean Cell Line Bank (KCLB, Seoul, Korea) and the RWPE-1 cells (CRL-11609) from ATCC; all the cells were grown in RPMI1640 supplemented with 10% FBS and 1% antibiotic (Welgene, Inc., Gyeongsan, Republic of Korea) at 37 °C in a humid environment with 5% CO_2_.

### 4.3. Cytotoxicity Assay

The cytotoxicity of Brassinin was assessed using a 3-(4,5-dimethylthiazol-2-yl)-2,5diphenyltetrazolium bromide (MTT) assay. In short, the PC-3, DU145, LNCaP, and RWPE-1 cells (1 × 10^4^ cells/well) distributed onto 96-well culture plates were exposed to various concentrations of Brassinin for 48 h. The cells were incubated with MTT (1 mg/mL) (St. Louis, MO, USA) for 2 h, followed by the addition of DMSO for 20 min. Then, the optical density (OD) was measured using a microplate reader (Molecular Devices, LLC, Sunnyvale, CA, USA) at 570 nm. The cell viability was determined as the percentage of viable cells in the Brassinin-treated group versus the untreated control.

### 4.4. Colony Formation Assay

The PC-3 cells were distributed onto 6-well plates at a density of 1000 cells per well for seven days. The colonies were fixed with glutaraldehyde (6.0% *v*/*v*), stained with crystal violet (0.5% *w*/*v*), and then counted by an inverted microscope.

### 4.5. Cell Cycle Analysis

The PC-3 cells (2 × 10^5^ cells/mL) were exposed to Brassinin (0, 60, and 80 μM) for 48 h. The cells were washed twice with cold PBS and fixed in 70% ethanol at −20 °C. The processed cells were incubated with RNase A (10 mg/mL) for 1 h at 37 °C and stained with propidium iodide (50 μg/mL) for 30 min at room temperature in the dark. The DNA contents of the stained cells were analyzed using FACSCalibur using the CellQuest Software.

### 4.6. Terminal Deoxynucleotidyl Transferase dUTP Nick end Labeling (TUNEL) Assay

The PC-3 cells (1 × 10^5^ cells/well) were plated onto 6-well plates and exposed to Brassinin (0 and 80 μM) at 37 °C for 48 h. The cells were fixed in 1% paraformaldehyde for 15 min and treated in 70% ethanol for 1 h. The cells were incubated with TUNEL buffer containing fluorescein isothiocyanate fluorescein-12-dUTP for 1 h at 37 °C and stained with propidium iodide (50 μg/mL) for 30 min at room temperature in dark. The DNA content of the stained cells was analyzed using FACSCalibur (Becton Dickinson, Franklin Lakes, NJ, USA) using the CellQuest Software.

### 4.7. Western Blotting

The PC-3, DU145, and LNCaP cells were exposed to various concentrations of Brassinin for 48 h, lysed with lysis solution (150 mM NaCl, 1% Triton X-100, 0.1% SDS, 1 mM EDTA 50 mM Tris–HCl, pH 7.4, 1 mM Na3VO4, 1 mM NaF, and 1 × protease inhibitor cocktail) containing protease inhibitors (Roche Diagnostics GmbH, Mannheim, Germany) and phosphatase inhibitors (SigmaAldrich, St. Louis, MO, USA) on ice, and centrifuged at 13,000 rpm for 20 min at 4 °C. The cell lysates were separated using sodium dodecyl sulfate polyacrylamide gel electrophoresis (SDS-PAGE) and transferred to a Hybond ECL transfer membrane. The membranes were blocked using 5% skim milk and incubated with primary antibodies for Caspase3, PARP, Bcl-2, Glut1, HK2, PKM2, LDH, SIRT1, β-catenin, c-Myc (Cell signaling Technology, Beverly, MA, USA), and β-actin (Sigma, St. Louis, MO, USA). The washed membranes were incubated with the corresponding horseradish peroxidase-conjugated anti-rabbit IgG (dilution 1:5000, cat. no. 7074; Cell Signaling Technology, Inc.) and anti-mouse IgG (dilution 1:5000, cat. no. 7076; Cell Signaling Technology, Danvers, MA, USA). The immunoreactive polypeptides were analyzed using an enhanced chemiluminescence (Thermo Scientific, Rochester, NY, USA).

### 4.8. Co-Immunoprecipitation

The PC-3 cells were lysed in lysis solution (0.1% SDS, 150 mM NaCl, 1% Triton X-100, 50 mM Tris–HCl, pH 7.4, 1 mM NaF, 1 mM EDTA, 1 mM Na3VO4, and 1 × protease inhibitor cocktail). Then, the cells were immunoprecipitated with SIRT1 or β-catenin, followed by an addition of Protein A/G sepharose beads (Santa Cruz Biotechnology, Santa Cruz, CA, USA). The final processed proteins were subjected to immunoblotting with the indicated antibodies.

### 4.9. Measurement of ROS Generation

The levels of ROS production were assessed using 2,7-Dichlorofluorescein diacetate (DCFH-DA, Sigma, USA). In the absence or presence of NAC, the PC-3 cells were treated with Brassinin for 48 h, followed by a 30 min incubation at 37 °C with 10 μM DCFH-DA. The ROS fluorescence intensity was measured using FACSCalibur (Becton Dickinson, Franklin Lakes, NJ, USA).

## 5. Conclusions

In our current work on exploring the underlying antitumor mechanism of Brassinin, Brassinin was found to significantly enhance the cytotoxicity and attenuate the number of colonies and expressions of pro-caspase 3, pro-PARP, and Bcl-2 in PC-3 cells compared to DU145 and LNCaP cells, and it increased the sub-G1 population and number of TUNEL-positive cells in the PC-3 cells. Of note, Brassinin suppressed the expressions of Glut1, HK2, PKM2, LDH, SIRT1, c-Myc, and β-catenin in the PC-3 cells. Additionally, Brassinin disrupted the binding of SIRT1 with β-catenin and increased ROS production, while NAC reversed the ability of Brassinin to alleviate the expressions of pro-PARP, pro-Caspase3, SIRT1, and β-catenin in the PC-3 cells. Overall, these findings highlight the evidence that Brassinin induces apoptotic and anti-Warburg effects via the ROS-mediated inhibition of SIRT1, c-Myc, β-catenin, and glycolysis proteins as a potent anticancer candidate.

## Figures and Tables

**Figure 1 ijms-24-13912-f001:**
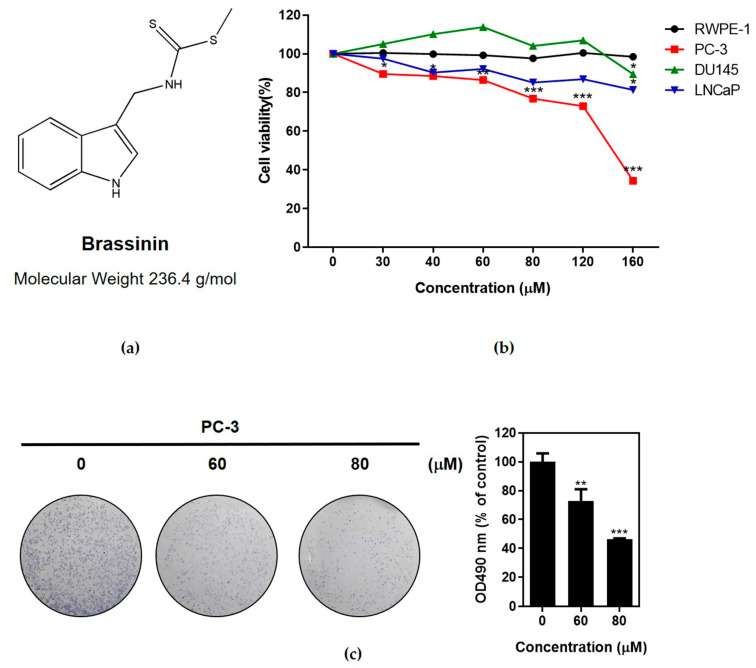
Effect of Brassinin on cytotoxicity and proliferation of prostate cancer cells. (**a**) Structure of Brassinin. Molecular weight = 236.4 g/mol. (**b**) Effect of Brassinin on cytotoxicity in PC-3, DU145, and LNCaP cells. These cells treated with various concentrations of Brassinin (0, 40, 60, and 80 μM) for 48 h were subjected to MTT assay. * *p* < 0.05, ** *p* < 0.01 and *** *p* < 0.001 versus untreated control. Data represent mean ± SD of three independent experiments. (**c**) Effect of Brassinin on the proliferation of PC-3 cells.

**Figure 2 ijms-24-13912-f002:**
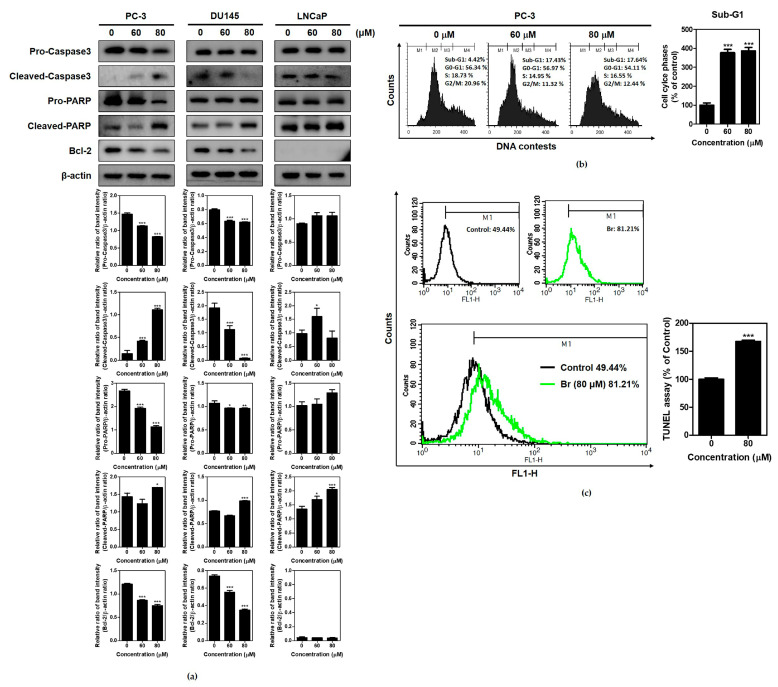
Effect of Brassinin on apoptosis in PC-3 cells. (**a**) Effect of Brassinin on apoptosis-related proteins in PC-3, DU145, and LNCaP cells. These cells were treated with Brassinin (0, 60, and 80 μM) for 48 h and cell lysates were subjected to Western blotting for Caspase3, PARP, and Bcl-2. (**b**) Effect of Brassinin on sub-G1 population in PC-3 cells. PC-3 cells were treated with Brassinin (0, 60, and 80 μM) for 48 h. The treated cells stained with propidium iodide (PI) were analyzed with flow cytometry. (**c**) Effect of Brassinin on TUNEL-positive cells in PC-3 cells via FACS analysis. Data are expressed as mean ± SD of three independent experiments. * *p* < 0.05, ** *p* < 0.01 and *** *p* < 0.001.

**Figure 3 ijms-24-13912-f003:**
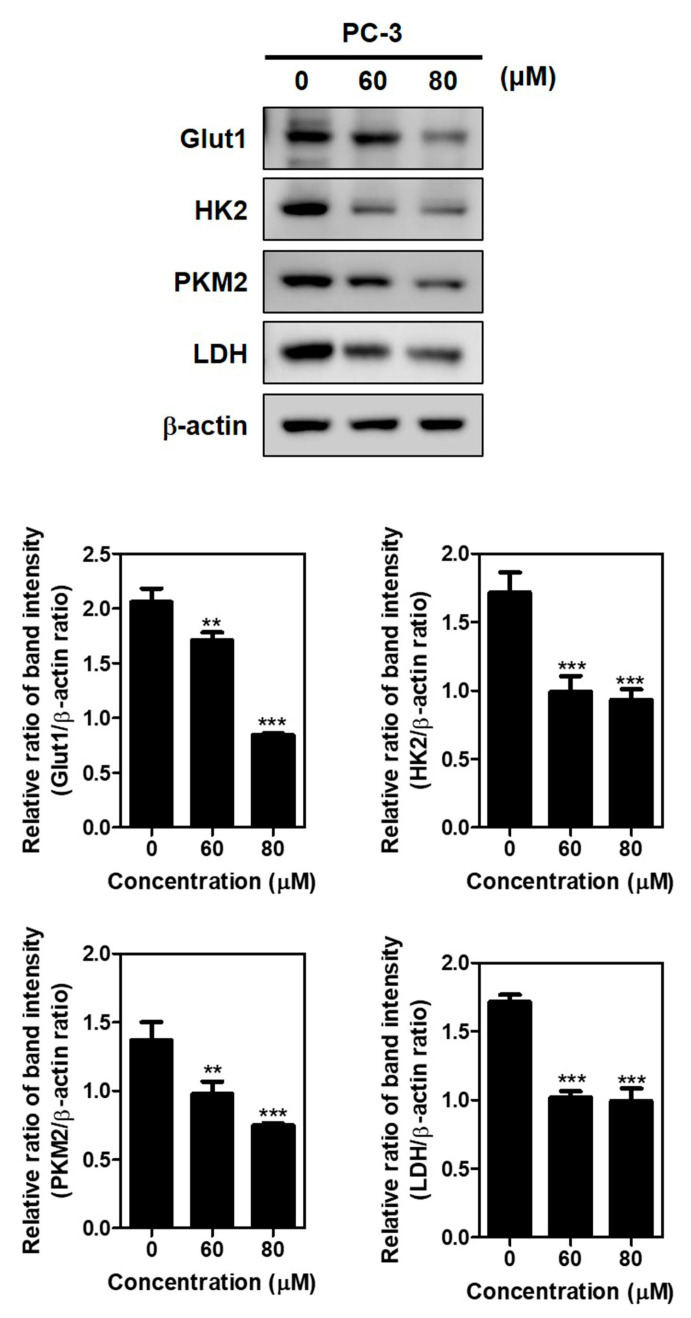
Effect of Brassinin on glycolysis-related proteins in PC-3 cells. PC-3 cells were treated with Brassinin (0, 60, and 80 μM) for 48 h. Cell lysates were subjected to Western blotting for Glut1, HK2, PKM2, and LDH in PC-3 cells. Data are expressed as mean ± SD of three independent experiments. ** *p* < 0.01 and *** *p* < 0.001 versus untreated control.

**Figure 4 ijms-24-13912-f004:**
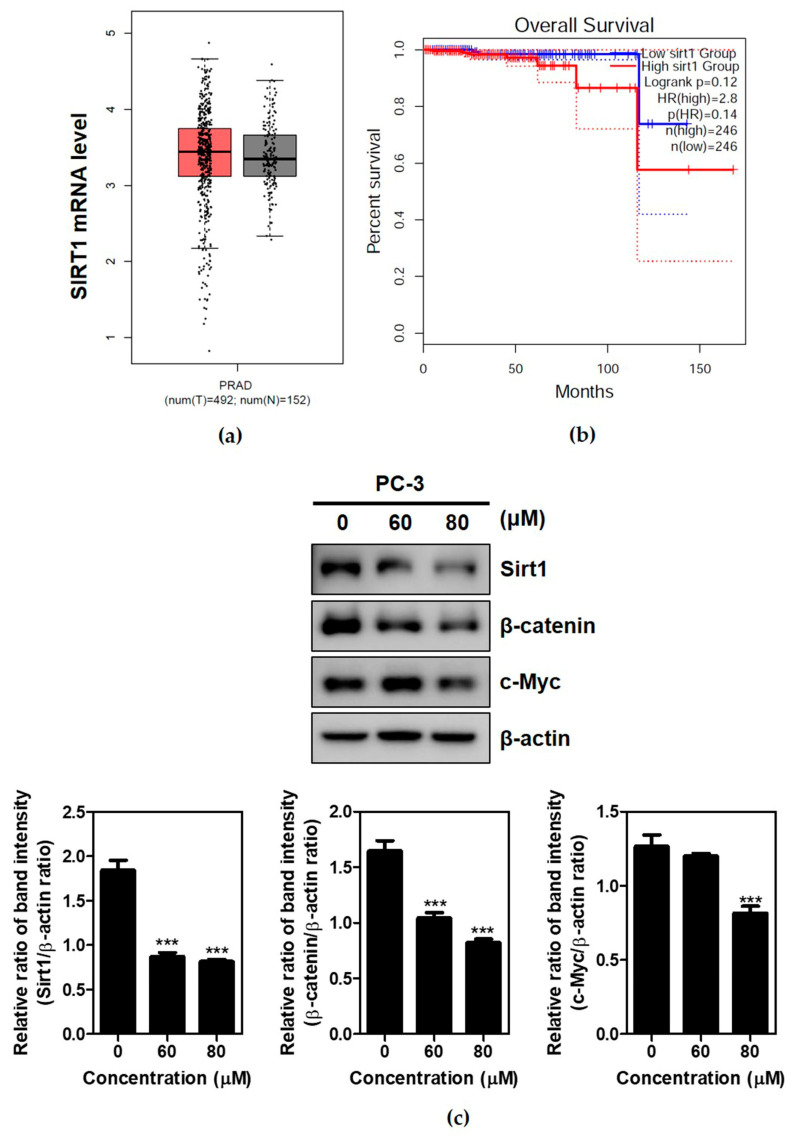
Effect of Brassinin on the expressions of SIRT1, β-catenin, and c-Myc in PC-3 cells. (**a**) The expression level of SIRT1 was upregulated in prostate cancers more than in normal prostate cancer, as determined with the GTEx database. (**b**) Kaplan–Meier survival rate in prostate cancer patients with low SIRT1 expression (n = 246, n = 246, HR = 2.8, *p* = 0.12). (**c**) Effect of Brassinin on SIRT1, β-catenin, and c-Myc expression in PC-3 cells via Western blotting. Data are expressed as mean ± SD of three independent experiments. *** *p* < 0.001; T(tumor group; red), N(normal group;grey).

**Figure 5 ijms-24-13912-f005:**
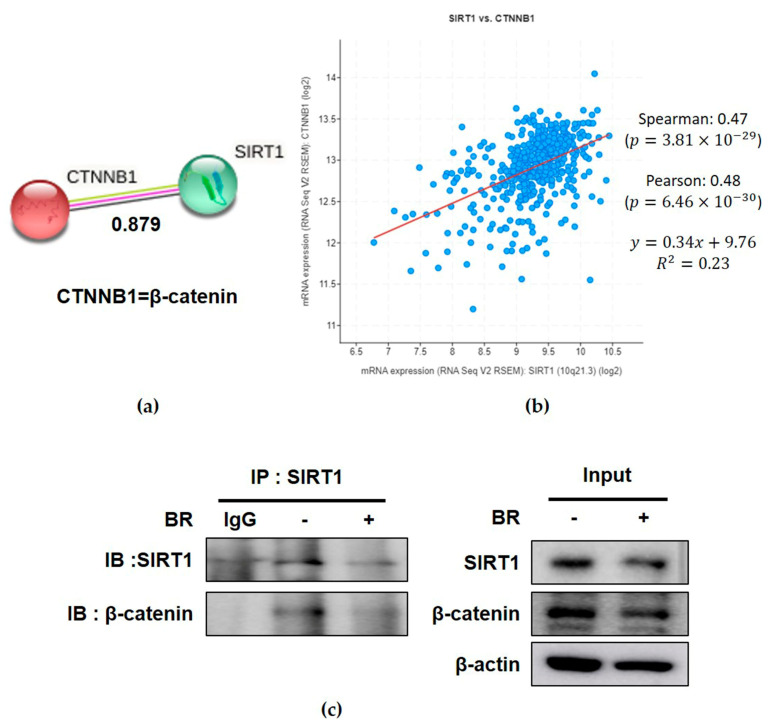
Effect of Brassinin on interaction between SIRT1 and β-catenin in PC-3 cells. (**a**) SIRT1 interacts with β-catenin according to the STRING database (interaction score: 0.879). (**b**) Correlation coefficient between SIRT1 and β-catenin at mRNA expression level in prostate cancer. (**c**) PC-3 cells were treated with 80 μM Brassinin for 48 h. Then, immunoprecipitation was performed with cell lysates from PC-3 cells using anti-SIRT1 antibody, and then Western blotting analysis was performed to detect SIRT1 and β-catenin in whole cell lysates.

**Figure 6 ijms-24-13912-f006:**
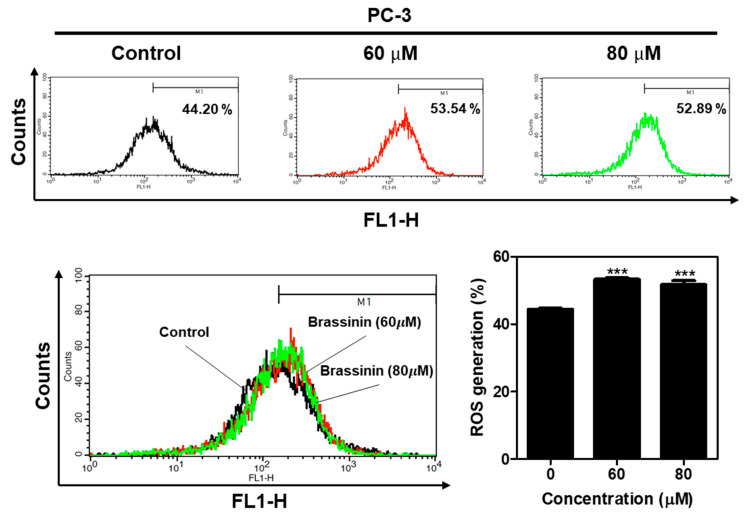
Effect of Brassinin on reactive oxygen species (ROS) generation in PC-3 cells. ROS production was measured in PC-3 cells treated with Brassinin (60 and 80 μM) for 48 h with DCFDA staining via FACSCalibur analysis. Data are expressed as mean ± SD of three independent experiments. *** *p* < 0.001 versus untreated control.

**Figure 7 ijms-24-13912-f007:**
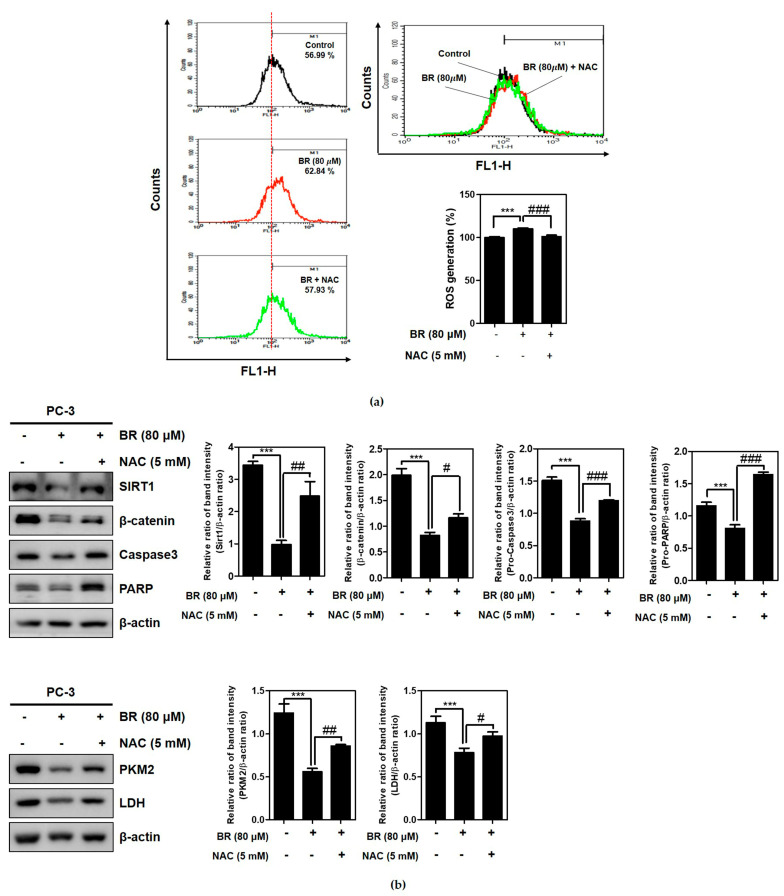
Effect of NAC on ROS generation. (**a**) Effect of N-Acetyl-L-cysteine (NAC) on ROS production in PC-3 cells via FACSCalibur. (**b**) Effect of NAC on the ability of Brassinin to abrogate the expressions of Sirt1, β-catenin, Caspase3, PARP, PKM2, and LDH in PC-3 cells. Data are expressed as mean ± SD of three independent experiments. *** *p* < 0.001 versus untreated control, # *p* < 0.05, ## *p* < 0.01, and ### *p* < 0.001 versus NAC- and Brassinin-treated group.

## Data Availability

All the data and materials supporting the conclusions are included in the main paper.

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
