# Peer review of "The Apoptotic and Anti-Warburg Effects of Brassinin in PC-3 Cells via Reactive Oxygen Species Production and the Inhibition of the c-Myc, SIRT1, and β-Catenin Signaling Axis"

_ijms, 2023, doi:10.3390/ijms241813912_

Round 1
Reviewer 1 Report (Previous Reviewer 2)
Manuscript is properly written and presented, however few modifications are still needed.

Moderate editing of English language required.
Author Response
Apoptotic and anti-Warburg effects of Brassinin in PC-3 cells via reactive oxygen species production and inhibition of c-Myc, SIRT1 and β-catenin signaling axis Overview The study highlights on the apoptotic mechanism and anti-Warburg effects involved in the presence of Brassinin in prostate (PC-3) cancer cell line. The study further elaborates the association of Brassinin and ROS in anticancer effect against PC-3 cancer cell line. The study conducted and reported is concise and significant.
Major Comments 1. Figure 1: (b) shows ~80% cell viability at 80 µM concentration, whereas (c) shows ~40%. Similar pattern at 60 µM concentration.
(Response) Sometimes short term cytotoxicity data are different from long term colony formation data. Herein long term colony formation data were better than short term cytotoxicity, which was added in Discussion
- Few figure representations are contradictory (Fig 6 and Fig 1b, Fig 2b). Authors stated “Brassinin significantly increased ROS generation in PC-3 cells”, yet unlike other results no concentration dependent activity was observed.
(Response) We absolutely agree with you regarding concentration dependency issue of Brassinin. However, we could accept concentration dependency of ROS production despite significant increase compared to untreated control. I can be expected 60 M as a threshold concentration of Brassinin for ROS generation.
- No discussion for Figure 6. Results of ROS at 80 µM concentration. Cytotoxicity is higher compared to 60 µM, yet ROS is lower.
(Response)Thanks for your critical comments. We discussed the results as you indicated. But there was no significant difference for ROS production between 60 and 80 µM of Brassinin.
- Please cite the reference articles for methodology adopted.
(Response) Thanks. Cited.
Minor Comments
- Please include statistical data regarding the prevalence and mortality of prostate cancer (https://gco.iarc.fr/).
(Response) Added in Introduction
- Figure 1a: “Weight 236.4”.
(Response) Corrected
- Line 229: Please specify the quantity of Brassinin dissolved in 0.1% DMSO.
(Response) Specified.
- Line 230: Please specify the storage conditions (Temperature).
(Response) Specified.
- Space should be given between values and units (e.g., 37 ºC (Line 252), 50 mM (Line 266)).
(Response) Corrected
- Please check the entire manuscript for the same. Remark The study has made valuable insights into the anticancer mechanism of Brassinin in PC-3 cancer cell line. However, certain modifications or improvements are still needed to further enhance the impact of the article. Thus, authors are requested to go through the stated suggestions and modify the manuscript accordingly.
(Response) Appreciated so much. Corrected
- Moderate editing of English language required.
(Response) Our MS was corrected by MDPI editing service.
{We certify that the following article Apoptotic and anti-Warburg effects of Brassinin in PC-3 cells via reactive oxygen species production and inhibition of c-Myc, SIRT1 and β-catenin signaling axis Sung-Hoon Kim has undergone English language editing by MDPI. The text has been checked for correct use of grammar and common technical terms, and edited to a level suitable for reporting research in a scholarly journal. MDPI uses experienced, native English speaking editors. Full details of the editing service can be found at ► https://www.mdpi.com/authors/english. Basel, Switzerland September 2023 english-7061}
Reviewer 2 Report (Previous Reviewer 1)
Comments:
Authors requires to clear the comments can be consider for publication are as follows;
1. All the abbreviation needs to elaborate at its initial usage, still found some abbreviation not elaborated like BAG1L.
2. Materials & Method section:- Authors needs to describe ROS assay method.
3. Introduction section:- Authors needs to reshape the introduction section with flow of conveying the purpose of conducting research in Prostate cancer, Brassinin role on controlling very deeply. This section requires lots of information relevant to the work with minimum 750 words.
4. Results section: Author performed many experiments with less description about the results and significance behind each results in detail. Interpretation between experiments with relativeness of result robustness.
Author needs to undergo extensive English editage service after clearing the comments.
Author Response
Authors requires to clear the comments can be consider for publication are as follows;
- All the abbreviation needs to elaborate at its initial usage, still found some abbreviation not elaborated like BAG1L.
(Response) BCL2-associated athanogene 1 L (BAG1L)
- Materials & Method section:- Authors needs to describe ROS assay method.
(Response) Added
- Introduction section:- Authors needs to reshape the introduction section with flow of conveying the purpose of conducting research in Prostate cancer, Brassinin role on controlling very deeply. This section requires lots of information relevant to the work with minimum 750 words.
(Response) Thanks.Added.
- Results section: Author performed many experiments with less description about the results and significance behind each results in detail. Interpretation between experiments with relativeness of result robustness.
(Response) Thanks. We described the results and the discussed the significance of the results data in Discussion based on your comments.
- Author needs to undergo extensive English editage service after clearing the comments.
(Response) Our MS was corrected by MDPI editing service.
{We certify that the following article Apoptotic and anti-Warburg effects of Brassinin in PC-3 cells via reactive oxygen species production and inhibition of c-Myc, SIRT1 and β-catenin signaling axis Sung-Hoon Kim has undergone English language editing by MDPI. The text has been checked for correct use of grammar and common technical terms, and edited to a level suitable for reporting research in a scholarly journal. MDPI uses experienced, native English speaking editors. Full details of the editing service can be found at ► https://www.mdpi.com/authors/english. Basel, Switzerland September 2023 english-7061}
This manuscript is a resubmission of an earlier submission. The following is a list of the peer review reports and author responses from that submission.
Round 1
Reviewer 1 Report

Required improvement.
Author Response
The manuscript (ijms-2518353) entitled “Apoptotic and anti-Warburg effects of Brassinin in PC-3 cells via reactive oxygen species production and inhibition of c-Myc, SIRT1 and β-catenin signaling axis” seems to be less novel, however, the authors studied the basic science in Prostrate cancer. The hypothesis proposed in this manuscript is clear with scientific conclusion, but author needs to improve manuscript much for potential publication. The experimental designs lack background explanation significance with signaling pathways, Warburg effect in comparison with others published work, and the writing must be improved to a higher degree relevant with correction and comments.
(Response) Thanks. Based on your kind comments we improved our MS.
The introduction and discussion section needs to improve much with strong scientific perspective.
(Response)Thanks. We added more detailed scientific perspectives in Introduction and Discussion.
The result concluded with a deficit in scientific justification for the apoptotic cell death happening through caspase3 & PARP, hence suggested to do the cleaved Cas3 & PARP experiment along with the confirmation experiment of DNA fragmentation will add advantage for the manuscript.
(Response) Thanks for your critical comments. Thus, we added cleavages of PARP and caspase 3 and TUNEL assay data for replacing DNA fragmentation.
.
Major& Minor comments: Required Graphical abstract.
(Response) Thanks. Figure 8 was modified as a graphical abstract.
Introduction 1st para; Required more discussion for why author selected only prostate cancer, any importance and serious requirements to conduct in prostate cancer.
(Response)Thanks for your kind and critical comments to improve our MS. We added your kind suggestion in Introduction.
Line 38; PSAM, EGFR, VEGF, and BAG1L expression, molecular signaling and expression significance requires to discuss more introduction section.
(Response)Thanks. Added.
Line 44; Significance of Warburg effect in cancer and specifically in Prostate cancer, discuss how it's effect going to affect in prostate/ cancer condition.
(Response) Thanks. Added.
Figure 1; Image of 1b shows 80% decrease in cell viability, but 1C& bar graph image shows 50% decrease in cell viability/ Proliferation of PC-3 cells. Please justify for better clarity. Figure legends; Statement of experiments conducted in triplicate, minimum procedure of treatment condition, cell seeded count for each experiment & statistical analysis needs to specify for each experiment is mandatory.
(Response) Thanks. Bar graphs were added for all figures and discussed.
Line 75; Required to interpret results more deeply and required intercorrelation interpretation between experiments. Figure 2; The expression of Pro-Cas3 & Pro-Parp assessed but required experiments for Cleaved Cas3 & Cleaved Parp evaluation also Bax level expression. for confirmation of PC-3&DU145 undergoing apoptosis pathway.
(Response) Thanks. Cleavages of Cas 3 and PARP were added.
Also, required table for SubG1, G1, S, G2/M phase cell cycle analysis (figure 2b) and discuss much about each phase. then claim for SubG1 is arrested.
(Response)Thanks. Each phase distribution Table was added and discussed.
Line 85; Required to interpret results more deeply relevant to Warburg's effect- glycolysis protein expression.
(Response)Thanks. Discussed.
Figure 6; Bar graph needs to represent in % of Control Ros generation.
(Response) Thanks. Quantified as % of Control Ros generation in Figure 6.
Figure 7; ROS level in Brassinin is not much higher than control, can author confirm this experiment once. And the LDH level for Brassinin (80uM) differs from the Glycolysis related protein (Figure 3).
(Response) Thanks. LDH blot was replaced.
Line 142; LNCaP cells didn't show prominent cytotoxicity, reframe the sentence. And Required discussion about the LNCap cells and their resistance than other two cell lines in discussion part. Line 168; Required reference study to cross verify the author's work on PPI of SIRT1/B-catenin
(Response)Thanks. Modified and discussed. Reference on SIRT1/B-catenin binding was added.
Reviewer 2 Report
Authors are requested to provide justifications or undertake the suggested comments.

Require moderate language corrections.
Author Response
Apoptotic and anti-Warburg effects of Brassinin in PC-3 cells via reactive oxygen species production and inhibition of c-Myc, SIRT1 and β-catenin signaling axis Overview The work reported shows the sensitivity of PC-3 (prostate cancer cell line) towards the Brassinin and followed by series of investigations to define the possible anticancer mechanism involved. The study is essential and warrants significance. Authors still need to provide clarifications or accommodate the following suggestions. Major corrections 1. Cell lines used were prostate cancer cells. Authors should at least perform cytotoxicity (MTT assay) for Brassinin with normal human cell lines (not cancer cell lines) as control. This will validate the Brassinin effect on PC-3 cell line.
(Response)The cytotoxicity on RWPE-1 was added.
What is the basis for limiting the concentration of Brassinin to 80 μM (only ~25% cell mortality was observed)? Authors could have increased the concentration (beyond 80 μM), which can reduce the time for incubation to 24 h.
(Response) Thanks. The cytotoxicity of Brassinin was also added at 120 and 160 μM in Figure1.
Figure 2a, 3a, 4b: Blots provided were not proper, most of the bands show blur pattern, please enhance the quality of the blots to show sharp bands for good visibility.
(Response) Thanks. Some blots were replaced.
Figure 4a: Authors stated, “higher expression of SIRT1 is observed in cancer patients”, but the figure shows minute difference (~0.2) compared to normal.
(Response) Thanks. Higher expression of SIRT1 is shown in cancer patients (n = 492) compared to normal (n = 152) with no significance (Figure 4a), lower SIRT1 expression was associated with improved overall survival in prostate cancer patients.
Figure 5c: Clear blot image is needed.
(Response) Thanks. Replaced.
Figure 2b and 6: 60 μM and 80 μM showed same effect. However, authors stated “Brassinin significantly increased ROS production at concentrations of 60 and 80 μM”.
(Response) Thanks for your critical comments. We removed “concentration dependent manner.”
Line 126: Authors stated “Brassinin increased ROS production in PC-3 cells in a concentration dependent manner”. However, Figure 7a shows only one concentration data (80 μM).
(Response)Sorry for making you confused. We removed “concentration dependent manner.”
Should provide a discussion with other metabolites (compounds) used for similar studies in same or different cancer cell lines. Literature is available. This will validate the results across different cancer cells.
(Response) Thanks. We added the similar data in previous published papers.
Minor corrections Line 14: “lung, and liver cancers. However, the underlying” Line 17: “Bcl-2” Line 32: Provide overall cancer statistics either World or China and possible contribution of prostate cancer to overall mortality [https://ourworldindata.org/cancer]. Line 47: “PKM2, and LDH” Line 48: “Recently, use of natural compounds to develop target” Line 50: “silibinin [22], and decursin [23]” Line 54 and 231: “anticancer mechanism” Figure 1b: “Asterix” not visible at 60 μM concentration. Place it below the line for visibility. Line 67: Remove the sentence “Cell viability was assessed by MTT assay” Line 68: “mean ± SD” Line 69: Place at the end of line 69. “Data represent mean ± SD. ** p
(Response) Corrected.
Reviewer 3 Report
It is tested in three different cell lines and Ic50 values are not reached in any of them. In fact, you only have some activity on a single line.
Author Response
The proposed article is interesting but lacks consistency. The introduction is quite concise and should be more specific and detailed.
(Response)Thanks. Details were added.
It is tested in three different cell lines and Ic50 values are not reached in any of them. In fact, you only have some activity on a single line.
(Response) We agree with you in this regard. However, we have found that Brassinin exerted apoptotic effect in PC-3 cells.
The western blot presented is not quantified and therefore the results shown cannot be assessed or affirmed. This happens with all the westerns that appear in the manuscript. Because it does not act in the same way in all cell lines and the results are not well quantified and present inconsistency, my decision is to reject it for this journal, which has such a high impact factor.
(Response) Thanks for your critical comments to improve the quality of our MS. We quantified all blots and performed further experiments based on the comments from three reviewers.
English can be improved but does not present serious errors.
(Response) Thanks for your kind comments. Here we carefully corrected English expression and careless flaws
Round 2
Reviewer 1 Report
Authors needs to improve manuscript much with scientific soundness and introduction, results and discussion section requires much improvement to get publish.
Author requires to rewrite manuscript.
Author Response
Respectful reviewer 1:
Many thanks for your critical and kind comments to improve the quality of out MS. We tried to rewrite our MS extensively in Introduction, Results and Discussion. Please check highlighed pargraphs. We sicerely hope your positive decision.
Best regards
sungkim
